# Patients with SARS-CoV-2-Induced Viral Sepsis Simultaneously Show Immune Activation, Impaired Immune Function and a Procoagulatory Disease State

**DOI:** 10.3390/vaccines11020435

**Published:** 2023-02-13

**Authors:** Andreas Limmer, Andrea Engler, Simone Kattner, Jonas Gregorius, Kevin Thomas Pattberg, Rebecca Schulz, Jansje Schwab, Johannes Roth, Thomas Vogl, Adalbert Krawczyk, Oliver Witzke, Gennadiy Zelinskyy, Ulf Dittmer, Thorsten Brenner, Marc Moritz Berger

**Affiliations:** 1Department of Anesthesiology and Intensive Care Medicine, University Hospital Essen, University of Duisburg-Essen, 45147 Essen, Germany; 2Department of Pediatric Cardiac Surgery, University Hospital Erlangen, 91054 Erlangen, Germany; 3Institute of Immunology, University of Münster, 48149 Münster, Germany; 4Department of Infectious Diseases, West German Centre of Infectious Diseases, University Hospital Essen, University of Duisburg-Essen, 45147 Essen, Germany; 5Institute of Virology, University Hospital Essen, University of Duisburg-Essen, 45147 Essen, Germany

**Keywords:** SARS-CoV-2, COVID-19, viral sepsis, bacterial sepsis, coagulopathy, immune activation, immune suppression, T cell, monocyte, neutrophil, damage associated molecular pattern (DAMP), mediator, biomarker

## Abstract

Background: It is widely accepted that SARS-CoV-2 causes a dysregulation of immune and coagulation processes. In severely affected patients, viral sepsis may result in life endangering multiple organ dysfunction. Furthermore, most therapies for COVID-19 patients target either the immune system or coagulation processes. As the exact mechanism causing SARS-CoV-2-induced morbidity and mortality was unknown, we started an in-depth analysis of immunologic and coagulation processes. Methods: 127 COVID-19 patients were treated at the University Hospital Essen, Germany, between May 2020 and February 2022. Patients were divided according to their maximum COVID-19 WHO ordinal severity score (WHO 0–10) into hospitalized patients with a non-severe course of disease (WHO 4–5, *n* = 52) and those with a severe course of disease (WHO 6–10, *n* = 75). Non-infected individuals served as healthy controls (WHO 0, *n* = 42). Blood was analyzed with respect to cell numbers, clotting factors, as well as pro- and anti-inflammatory mediators in plasma. As functional parameters, phagocytosis and inflammatory responses to LPS and antigen-specific stimulation were determined in monocytes, granulocytes, and T cells using flow cytometry. Findings: In the present study, immune and coagulation systems were analyzed simultaneously. Interestingly, many severe COVID-19 patients showed an upregulation of pro-inflammatory mediators and at the same time clear signs of immunosuppression. Furthermore, severe COVID-19 patients not only exhibited a disturbed immune system, but in addition showed a pronounced pro-coagulation phenotype with impaired fibrinolysis. Therefore, our study adds another puzzle piece to the already complex picture of COVID-19 pathology implying that therapies in COVID-19 must be individualized. Conclusion: Despite years of research, COVID-19 has not been understood completely and still no therapies exist, fitting all requirements and phases of COVID-19 disease. This observation is highly reminiscent to sepsis. Research in sepsis has been going on for decades, while the disease is still not completely understood and therapies fitting all patients are lacking as well. In both septic and COVID-19 patients, immune activation can be accompanied by immune paralysis, complicating therapeutic intervention. Accordingly, therapies that lower immune activation may cause detrimental effects in patients, who are immune paralyzed by viral infections or sepsis. We therefore suggest individualizing therapies and to broaden the spectrum of immunological parameters analyzed before therapy. Only if the immune status of a patient is understood, can a therapeutic intervention be successful.

## 1. Introduction

Severe acute respiratory syndrome coronavirus type 2 (SARS-CoV-2) infections cause a mild disease in most individuals. However, about 10–20% of coronavirus disease 2019 (COVID-19) patients progress to a severe course of disease with pneumonia and respiratory failure [1]. If an acute respiratory distress syndrome (ARDS) develops it occurs on average around 8–9 days after symptom onset and is the cause of death in up to 70% of COVID-19 patients [1]. Thromboembolism is another prominent clinical feature of COVID-19 and 5–30% of in-hospital patients develop a clinically apparent thrombotic event [2]. Furthermore, up to 80% of severe COVID-19 patients fulfil sepsis-3 criteria caused by a dysregulated immune response to infection [3,4]. Because of the dysregulated immune and coagulation systems these patients may suffer from organ damage as well as from secondary bacterial or fungal superinfections [4,5,6,7].

In the mild and early stages of SARS-CoV-2 infection, the virus remains confined to the upper respiratory tract, stimulating a low level of innate immune response. Within 5–6 days of symptom onset, SARS-CoV-2 viral load reaches its peak, and the virus may spread to the lung. At this stage of the disease, an optimal but controlled innate and adaptive immune response will help to control infection. Successful elimination of the virus from recovered patients depends on the presence of an adequate number of adaptive immune cells, of immune regulatory molecules, and of neutralizing antibodies [1]. These processes lead to viral clearance and only minimal lung damage, ultimately resulting in recovery.

However, in some cases, a dysfunctional immune response occurs, which may cause severe lung and even systemic pathology. In case of a compromised immune response, uncontrolled cytokine production (“cytokine storm”) may lead to further accumulation of immune cells in the lungs, causing overproduction of pro-inflammatory cytokines damaging the lung microstructure. The resulting “cytokine storm” eventually extends to other organs, leading to multiple organ damage. Dysregulated immune responses have been described in SARS-CoV-2 infections on various levels [7,8].

Besides many reports of immune-mediated organ damage due to SARS-CoV-2 infection, a few reports demonstrate an impairment of the innate and adaptive immunity, reflected by a decrease of immune cell numbers, down-modulating of HLA-DR on monocytes, an increased number of immature granulocytes as well as strongly reduced reactivity of T cells and monocytes [9,10,11,12]. However, the prevailing paradigm that has guided the therapeutic approach to COVID-19 is, that patients die from the effects of a cytokine storm-mediated (hyper-) inflammation with organ injury of e.g., the lung [4,5,13,14,15,16].

As the inflammation and coagulation systems are highly connected, an increasing number of studies during the pandemic has shown that the observed organ dysfunctions in COVID-19 patients were caused by immune responses, eventually causing dysregulation of the coagulation system [6,17,18,19]. The hyper-coagulable disease state with a frequent occurrence of venous thromboembolic events (VTE) and pulmonary in situ thrombosis in COVID-19 patients was recently termed immune-thrombosis [19]. It is assumed, that SARS-CoV-2 causes an activation of leukocytes, which interact with platelets and coagulation factors leading to intravascular clot formation and micro-thrombotic complications in the lungs and other organs. By so far unknown mechanisms, both inflammatory and coagulation cascades are not controlled, but rather stimulated in a self-stimulatory positive feedback loop [19,20].

As the mechanisms causing dysregulation of immune and coagulation processes in COVID-19 patients are still not understood, we performed broad spectrum analyzes of phenotypic and functional immune parameters as well as of pro- and anti-coagulation factors in plasma of non-severely and severely sick COVID-19 patients and in healthy controls.

## 2. Methods

### 2.1. Study Population

In the here presented study, three groups were analyzed based on the World Health Organization (WHO) 11-point ordinal severity scale reflecting severity of COVID-19 symptoms [21]. The WHO scale scores range from 0 to 10, with 0 points for uninfected individuals and 10 points reflecting death. The control group consisted of healthy individuals with no clinical or virological evidence of infection (WHO score 0). The COVID-19 patients were included from May 2020 until March 2022 and divided into two groups, i.e., hospitalized severe (severe COVID-19) and less severe (non-severe COVID-19) patients, corresponding to WHO scores 6–10 and 4–5, respectively [21].

### 2.2. Ethics

Prior to the study each participant provided informed written consent. The study was approved by the Ethics Committee of the Medical Faculty of the University of Duisburg/Essen (#20-9216-BO and #17- 7824-BO amendment).

### 2.3. Cellular Analyzes

Blood containing anticoagulants (ethylenediamine tetra acetic acid (EDTA), heparin) or without anticoagulants (serum) was collected within 24 h (day 0) and if possible, seven days (day 7) after admission to the hospital. Immune cells in blood were analyzed immediately (flow cytometry, cell culture), while parameters in plasma or serum were analyzed after preservation at −80 °C (multiplex enzyme linked immunosorbent assay, ELISA). Absolute numbers of white blood cells (WBC), lymphocytes, monocytes, granulocytes, and thrombocytes (platelets) were determined by using a Sysmex XP300 Automated Hematology Analyzer (Sysmex, Norderstedt, Germany).

### 2.4. Flow Cytometry

As described previously [22], 50 μL of blood were stained with a cocktail of fluorochrome-labelled antibodies recognizing surface molecules of immune cells and measured with a CytoFLEX S (Beckman Coulter, Krefeld, Germany): CD45-PerCPCy5.5 (WBC), CD14-PECy7 (monocytes), CD15-APCFire750 (granulocytes), HLA-DR-BV421 (MHC class II), CD2-BV605 (lymphocytes), CD3-AF488 (T cells), CD11b-APC (integrin, MAC-1). All antibodies and reagents needed for staining were purchased from BioLegend (Koblenz, Germany).

### 2.5. Multiplex ELISA

All metabolites were measured in platelet-free plasma using LegendPlex panels from BioLegend: Human Inflammation Panel 1 (IL-1β, IFN-α2, IFNγ, TNFα, MCP-1 (CCL2), IL-6, IL-8 (CXCL8), IL- 10, IL-12p70, IL-17A, IL-18, IL-23, and IL-33), Human Inflammation Panel II (sTREM-1, PTX3, sCD40L, sCD25 (IL-2Ra), CXCL12 (SDF-1), sST2(IL-33R), sTNF-RI, sTNF-RII, sRAGE), Human Vascular Inflammation Panel I (myoglobin, MRP8/14, NGAL, CRP, MMP-2, OPN, MPO, SAA, IGFBP-4, ICAM-1, VCAM-1, MMP-9, cystatin), Human Thrombosis Panel (IL-6; *p*-selectin, D-Dimer, PSGL-1, tPA, CD40L, PAI-1, tissue factor, Factor IX, CXCL8 (IL-8)) and Human Fibrinolysis Panel (fibrinogen, antithrombin, plasminogen, prothrombin, Factor XIII; Human TH1/2 Panel (IL-2, IL-4, IL- 5, IL-6, IL-10, IL-13, IFNγ and TNFα). Multiplex ELISAs were performed following manufacturer’s instructions. In short, cell-free plasma was incubated with beads, followed by incubation with a cocktail of PE-labelled secondary antibodies and measured with a CytoFLEX S (Beckman Coulter). The MRP8/14 concentrations in plasma samples were quantified using an in-house sandwich ELISA, which detects both active heterodimers and inactive tetramers formed by MRP8 and MRP14 [23].

### 2.6. Functional Immunoassays–Stimulation of Monocytes and T Cells

A 50 μL volume of blood was incubated for 5 h with LPS (50 ng/m; Invivogen, *S. minnesota* R595) and 30 μL of beads (anti-biotin MACSiBeads, concentrations were not disclosed; Miltenyi Biotec (Bergischgladbach, Germany) conjugated with antibodies against human CD2, CD3, and CD28 (all from BioLegend). The supernatant was collected and frozen at −80 °C until measured in multiplex ELISA (LegendPlex Human Th1/2 Panel). Raw data from functional immunoassays were normalized to 100,000 monocytes and T cells, respectively, by comparing absolute cell numbers acquired by flow cytometry analysis. Analysis of flow cytometric data was performed by using FCS Express 6 software from DeNovo (Research Edition, Glendale, CA, USA). Raw data from multiplex ELISAs were analyzed with the LEGENDplex data analysis software provided by the manufacturer.

### 2.7. Phagocytosis Assay (E. coli-pHrodo)

A 50 μL volume of blood was incubated for 1 h with 10 μL of *E. coli*-pHrodo (Fisher Scientific, 1 mg/mL). After incubation, uptake of *E. coli*-pHrodo into acidified vesicles of monocytes (CD45^+^CD14^+^) and granulocytes (CD45^+^CD15^+^) was determined by flow cytometry. Besides uptake of *E. coli*-pHrodo, upregulation of CD11b on monocytes and granulocytes was measured.

### 2.8. Isolation and Quantification of Mitochondrial DNA

Total DNA from plasma was isolated using the QIAamp DNA Blood Mini Kit (Qiagen, Hilden, Germany) according to the manufacturer’s instructions. For quantification of mitochondrial DNA (mtDNA) Real-time quantitative PCR was performed on a StepOne Plus PCR System (Applied Biosystems, Darmstadt, Germany) using the Takyon ROX SYBR MasterMix dTTP Blue (Eurogentec, Seraing, Belgium) and the following cycling conditions: 5 min at 95 °C followed by 40 cycles of 3 s at 95 °C and 40 s at 60 °C. A meltcurve was performed after each run. The primers for ATPase 6 (5’-TCCCCATACTAGTTATTATCGAAACCA-3′ and 5′-GCCTGCAGTAATGTTAGCGGTTA-3′) and D-Loop (5′-TGCACGCGATAGCATTGC-3′ and 5′-AGGCAGGAATCAAAGACAGATACTG-3′) were designed with Primer Express 3.0 Software (Applied Biosystems). For absolute quantification of mitochondrial DNA, serially diluted DNA samples generated from purified specific PCR products (Invisorb Fragment CleanUp Kit, Stratec Molecular, Germany) were used as external standards in each run.

### 2.9. Data and Statistical Analysis

Data are presented as median with interquartile range unless indicated otherwise. Data analysis was performed using GraphPad Prism 8 (GraphPad Software Inc., San Diego, CA, USA). Data normality was tested using the Shapiro–Wilk Test. For comparison of all three study groups (healthy controls, non-severe COVID-19, severe COVID-19) the Kruskal–Wallis test was performed. To analyze parameters from COVID-19 patients from different time points (day 0 and 7) a 2-way ANOVA or, in case of missing values, a mixed-effects analysis was performed. The level of significance was set at *p* < 0.05.

## 3. Results

### 3.1. Study Population

Demographic data of the COVID-19 patients are summarized in Table 1. Altogether, 127 COVID-19 patients and 42 healthy controls were included. The COVID-19 patients were hospitalized and further divided into patients outside the intensive care unit (ICU) corresponding to a WHO scale of 4–5 (non-severe COVID-19; *n* = 52) and COVID-19 patients in the ICU suffering from severe symptoms (severe COVID-19; *n* = 75) corresponding to a WHO scale of 6–10.

### 3.2. Severe COVID-19 Patients Show Increased Granulocyte but Decreased Lymphocyte Numbers

In a first analysis, we determined the number of characteristic immune cells, such as white blood cells (WBCs) and lymphocytes in general and cellular subtypes such as granulocytes, monocytes, NK cells and T cells. Figure 1 shows absolute numbers of major immune cell populations determined by flow cytometry. Similar to other reports, we observed that the number of WBCs in severe COVID-19 patients increased (Figure 1a), most likely due to a numeric increase of granulocytes (Figure 1b). Contrariwise, the number of lymphocytes in general and T cells and NK cells specifically, decreased in severe COVID-19 patients (Figure 1c–f). While the numbers of monocytes did not change between groups (Figure 1f), platelet numbers were only significantly different between healthy controls and the group of non-severe COVID-19 patients (Figure 1g).

### 3.3. Higher Concentrations of Pro- and Anti-Inflammatory Mediators in Plasma of Severe COVID-19 Patients

Figure 2 as well as Appendix A show a broad selection of various well-known pro- and anti-inflammatory cytokines in the plasma. Of the analyzed pro-inflammatory cytokines, the concentrations of interleukin (IL)-1β, monocyte chemotactic protein 1 (MCP-1), IL-6, IL-8 and IL-17A were significantly higher in severe COVID-19 patients compared to healthy controls (Figure 2a–e). The concentration of IL-12p70 was significantly elevated in severe COVID-19 patients compared to non-severe COVID-19 patients (Figure 2f).

### 3.4. Higher Concentrations of Pro- and Anti-Inflammatory Soluble Receptors in Plasma of Severe COVID-19 Patients

When analyzing plasma levels of soluble membrane-bound cytokine receptors and chemokines, we observed an association with disease severity in COVID-19 patients (Figure 3). Concentrations of soluble receptors, such as triggering receptor expressed on myeloid cells 1 (sTREM-1), receptor for advanced glycation end products (sRAGE), sCD25, suppressor of T cell receptor signaling/soluble interleukin 1 receptor-like 1 (sST2/IL-33R), TNF receptor-I (sTNFR-I), sTNFR-II, as well as CX3CL1 and CXCL12 were highest in severe COVID-19 patients compared to both, healthy controls and non-severe COVID-19 patients.

### 3.5. Biomarkers Reflecting Organ Dysfunction Are Significantly Upregulated in Severe COVID-19 Patients

When looking at mediators reflecting organ dysfunction, the most severe differences were observed in severe COVID-19 patients (Figure 4). Biomarkers indicating oxidative stress (myeloperoxidase (MPO), Figure 4a) and overall organ stress (myoglobin, MMP-9, SAA; Figure 4b–d) as well as organ-specific markers for kidney (neutrophil gelatinase-associated lipocalin (NGAL), Cystatin C; Figure 4e,f) and endothelial (ICAM-1, VCAM-1; Figure 4g,h) damage were significantly increased in COVID-19 patients in general, but most pronounced in severe COVID-19 patients.

### 3.6. Concentrations of Soluble Pattern Recognition Receptors Are Increased in COVID Patients but Strongly Increased in Plasma of Severe COVID-19 Patients

Next, we analyzed the expression of acute phase proteins, damage associated molecular patterns (DAMPs), and scavenger receptors in plasma samples. As shown in Figure 5, the concentration of DAMPs was significantly increased in COVID-19 patients, with the highest increases in severe COVID-19 patients. While mtDNA was significantly but only moderately elevated (Figure 5a,b), the concentration of the calprotectin MRP8/14 was strongly increased (Figure 5c). Because of inflammation, the long pentraxin PTX3 was upregulated in COVID-19 patients in general, but much more pronounced in severe COVID-19 patients (Figure 5d).

### 3.7. Impaired Immune Function of T Cells and Monocytes in Severe COVID-19 Patients

To determine the immune status of COVID-19 patients, we measured HLA-DR expression on monocytes and granulocytes as a generally accepted indicator of immunosuppression. In addition, we determined the functional immune status of immune cells present in blood. We observed a selective downregulation of HLA-DR on monocytes of severe COVID-19 patients, while HLA-DR on neutrophils remained unchanged throughout all groups (Figure 6a,b). The ability of monocytes and T cells to respond to the stimulation with cell-specific triggers (e.g., LPS, anti-CD3/CD2/CD28) was dampened most significantly in severe COVID-19 of monocytes with LPS, secretion of pro-inflammatory cytokines IL-6, TNFαand IFNFγ (Figure 6c–g) was significantly reduced, while stimulation of T cells with antiCD3/CD2/CD28 resulted in reduced levels of IL-2, IL-6, TNFαand IFNγ (Figure 6h–l). Overall, cytokine production after stimulation was decreased in both monocytes and T cells of COVID-19 patients as compared to controls. The changes were most pronounced in severe COVID-19 patients implying a more severe immunosuppression in these patients.

### 3.8. Impaired Phagocytic Activity of Monocytes and Granulocytes in Severe COVID-19 Patients

Antigen uptake (*E. coli*-pHrodo) and upregulation of the integrin CD11b on monocytes and granulocytes represent clear signs of functional cellular activity. Comparing COVID-19 patients and controls shows that the uptake of the fluorescent *E. coli*-derived particle pHrodo (*E. coli*-pHrodo) was only reduced in monocytes of severe COVID-19 patients, while this function remained unchanged in non-severe COVID-19 patients (Figure 7a). In granulocytes, the uptake of *E. coli*-pHrodo was most severely decreased in severe COVID-19 patients, showing that phagocytosis was significantly impeded in patients suffering from severe consequences of SARS-CoV-2 infection (Figure 7b). Interestingly, activation of monocytes and granulocytes following stimulation with *E. coli* particles was reduced in COVID-19 patients as measured by the upregulation of CD11b (Figure 7c,d).

### 3.9. Increased Coagulation and Decreased Fibrinolysis in Severe COVID-19 Patients

Next, we analyzed changes in the coagulation status of COVID-19 patients compared to healthy controls. The coagulation factors, fibrinogen, D-Dimer, tissue factor, P-Selectin, factor IX, and P-selectin glycoprotein ligand-1 (PSGL-1) were upregulated while factor XIII was reduced in blood of both groups of COVID-19 patients (Figure 8a–g). Compared to healthy controls prothrombin was elevated in non-severe COVID-19 patients but not in severe COVID-19 patients (Figure 8h). Furthermore, plasminogen- activator-inhibitor type 1 (PAI-1), a strong inhibitor of fibrinolysis, was upregulated (Figure 8i), while plasminogen and tissue plasmin-activator (tPA) concentrations were unaffected (Figure 8j,k), implying that fibrinolysis was impeded in both groups of COVID-19 patients. In addition, no compensatory anti-coagulation reaction could be observed, as the level of e.g., anti-thrombin (AT) III did not differ significantly between healthy controls and COVID-19 patients (Figure 8l).

### 3.10. Only Minor Phenotypic and Functional Changes over Time in COVID-19 Patients

Data presented so far describe the clinical situation of COVID-19 patients at admission to the hospital (day 0) in comparison to healthy controls, whereas data on phenotypical and functional parameters at day 7 were also available. Overall, the dichotomy between severe COVID-19 and non-severe COVID-19 patients described at day 0 remained unchanged at day 7. However, some parameters changed over time. For example, the concentrations of IL-6 and IL-10 were higher on day 7 in severe COVID-19 patients compared to concentrations measured at onset (Appendix A).

## 4. Discussion

Viral infections can cause extensive pathology either by an overwhelming uncontrolled immune response (“cytokine storms”) or by a dysregulated immune response not able to control the infection. Meanwhile, many reports exist which characterize COVID-19 patients on a molecular level. Overall, these reports demonstrate that COVID-19 patients suffer from an immune mediated pathology [7,8,11,24,25]. Accordingly, up to 80% of intensive care COVID-19 patients fulfil sepsis-3 criteria with strong signs of organ dysfunction and coagulopathy, which is—by definition—caused by a dysregulated immune response to infection [3,4,6,18,26,27].

The COVID-19 patients in our study were divided into two groups, hospitalized severe and non-severe COVID-19 patients, and were compared to healthy controls. With increasing severity, COVID-19 patients showed elevated levels of inflammatory cytokines in blood accompanied by clear signs of a functional immunosuppression. In this respect, monocytes and T cells were functionally impaired showing a reduced cytokine production after stimulation, while monocytes and granulocytes exhibited a diminished capacity to phagocytose bacterial structures. Furthermore, the concentration of soluble receptors in blood, a parameter often found in severely ill patients, was also significantly elevated. In addition, and in line with previous reports [9,10,28,29], our study provides evidence that COVID-19 patients with viral sepsis suffer from a distinct immune disturbance accompanied by a severe COVID-19 associated coagulopathy (CAC). Most importantly, the data presented in this study show, that severe COVID-19 patients suffer from a strong immunosuppression while the pro-inflammatory immune response is still ongoing.

Especially when analyzing functional parameters, such as phagocytosis and recognition of bacterial structures, we observed a significantly reduced immune response in severely ill COVID-19 patients as compared to the other two groups, i.e., non-severe COVID-19 patients and healthy controls. Both, production of pro-inflammatory cytokines after stimulation with LPS and phagocytosis of *E. coli* (*E. coli*-pHrodo) were negatively affected. The immunosuppression was not fully developed, as monocytes and granulocytes were able to mount a reduced but detectable immune response.

Our findings show that not only the innate, but also the adaptive immunity is impaired. When stimulating T cells with anti CD3/CD2/CD28 beads, we observed a pronounced reduction of cytokine production, especially of TH1 cytokines known to stimulate cellular immune responses (IL-2, IFNγ, TNFα, IL-6). These findings support the notion that severely sick COVID-19 patients indeed are immune compromised similar to patients with bacterial sepsis [12,15]. Our data are in line with a study comparing patients with COVID-19, bacterial sepsis, and critically ill patients without sepsis [12,15]. As in our study, the authors described a strong reduction of T cell numbers and reactivity as well as a down-modulation of HLA-DR on monocytes in COVID-19 and septic patients [12,15]. As the expression of soluble receptors (sCD25, sRAGE, sTNFR I/II, PTX3) can also be found in patients with bacterial sepsis, our findings and those of others support the notion that pathologies in COVID-19-induced viral sepsis and bacterial sepsis share a similar mechanistic origin. Further supporting the similarity between bacterial sepsis and COVID-19-induced viral sepsis, we observed an elevation of factors associated with oxidative stress (e.g., MPO) and factors released from damaged (organ) tissue (NGAL, Cystatin C, myoglobin). These data support the notion that organ dysfunction is the main criterion of sepsis independent of the underlying pathogen [3].

Initial findings suggested that the pathology caused by SARS-CoV-2 is the result of a cytokine storm with pro-inflammatory cytokines (e.g., IL-6 being the most important cytokine) [13]. Similar to these studies, we also observed a significant increase in IL-6 in the plasma of severe COVID-19 and non-severe COVID-19 patients. However, despite a significant increase in pro-inflammatory cytokines we would not call this response a cytokine storm. Our notion is supported by recent publications, in which the authors did not find signs of a cytokine storm in COVID-19 patients, despite a clear pro-inflammatory response [11,30,31,32,33,34,35]. Rather than a cytokine storm, the authors found strong signs of an immunosuppression [12,14,15,24,25]. However, why did anti-inflammatory therapies (anti-IL-6, glucocorticoids) improve survival of COVID-19 patients, if there was no cytokine storm? We and other groups speculate that the success of an anti-inflammatory treatment might depend on the immune status of the individual patient [12,14,15,24,25]. It is conceivable that patients who do not (yet) exhibit signs of an immunosuppression might benefit from an anti-inflammatory therapy, while for immunocompromised patients the same therapy might be detrimental [36]. In the study by Remy et al., administration of recombinant IL-7 could partially restore immune function, suggesting that some COVID-19 patients require immune stimulation rather than immunosuppression [15]. This is in line with current clinical trials, in which immune stimulatory therapies (e.g., with IL-7, GM-CSF, check-point inhibitors) were beneficial for patients with bacterial sepsis [36,37]. Additionally, one must be aware of the fact that e.g., IL-6 itself is neither a pro-, nor an anti-inflammatory cytokine [38].

Our study shows that COVID-19 patients suffer from a COVID-19 associated coagulopathy. The COVID-19 patients exhibited a clear pro-coagulatory phenotype, as evidenced by elevated levels of D-dimer, fibrinogen, P-Selectine, and PSGL-1. While clot formation was accelerated, at the same time fibrinolysis was inhibited, as evidenced by an elevated concentration of PAI-1 and an unaffected concentration of plasminogen. Furthermore, at the same time we did not observe a compensatory counter-regulation (e.g., by ATIII). Our findings are in line with various publications demonstrating that COVID-19 patients in general suffer from CAC [17,20,34,39]. There is also increasing evidence that disturbance of the coagulation system differs in bacterial and in COVID-19 viral sepsis. Patients suffering from bacterial sepsis show early signs of thrombocytopenia and quickly develop disseminated intravascular coagulation (DIC), a highly pathologic condition, where coagulation factors become limited due to over-activation. If DIC is induced during bacterial sepsis, it is called sepsis induced coagulopathy (SIC).

However, in most severe COVID-19 patients, the number of thrombocytes is not changed and only a few patients develop DIC [19,20,39,40,41,42,43]. An interesting aspect has been offered by Nossent et al., who found that dysregulation of coagulation was much more pronounced locally in the lung than systemically in the blood [20]. The authors suggest that in severe COVID-19 patients a local pulmonary rather than a systemic pro-coagulant and inflammatory cytokine storm prevails.

We have previously shown that in systemic bacterial infections, PAMPs and DAMPs can cause immunosuppression [21,44,45]. As in COVID-19 patients, immunosuppression seems to be induced by the virus rather than by bacteria, we were wondering whether SARS-CoV-2 could have induced the expression of endogenous TLR-ligands known to suppress immune responses. In this respect, we and others have previously published that mtDNA is a highly selective marker for sepsis and that in a mouse model mtDNA has been shown to suppress adaptive immunity against secondary viral infections [22,46,47]. Besides mtDNA, calprotectins, also called MRP8/14 and S100A8/9, have been shown to be highly expressed in patients with bacterial sepsis. In mice, similar to LPS, MRP8/14 can induce septic shock via TLR4 [48,49]. As we have shown that bacteria and TLR-ligands such as LPS can induce immune paralysis [50], we speculate that MRP8/14 can induce immune suppression in COVID-19 patients as well. This is in line with recent publications showing that MRP8/14 can induce tolerance especially in the myeloid compartment [44,45,51]. It is not understood why viral infections can also cause sepsis although viruses do not express ligands for TLR-4 or TLR-2, the prototypic TLRs involved in bacterial sepsis and septic shock. It is interesting to speculate that the induction of endogenous TLR-ligands (DAMPs) and especially calprotectin (MRP8/14), can provide a link, as calprotectin can cause septic shock similar to LPS by binding to TLR4 [48,49].

Over the last years many clinical and biochemical markers have been identified, which were shown to be able to differentiate between disease severity and to predict mortality [36,52,53]. While the demographic markers “age” and “sex” were useful predictors in many studies, the ratio of neutrophil and lymphocyte numbers (NLR) has been shown to be the most reliable marker available [53]. In our study, NLR was highly significant (AUC = 0.848), while the demographic markers “age” and “sex” were not significant and therefore unable to predict differences in severity between COVID-19 patients (Table 1). Calprotectin, found in blood of COVID-19 patients, has been reported to correlate with the severity of disease [54,55] and thus could function as a biomarker. Our study supports the use of calprotectin as prognostic marker as ROC analysis gave an AUC value of 0.736 when comparing calprotectin concentrations in severe and non-severe COVID-19 patients.

Although consistent conclusions can be drawn from our study, there are some limitations that need to be addressed. The number of patients and time points of analysis is low, limiting the generalizability of the results. A further limitation is the coverage of different viral strains causing COVID-19. It is not known whether all strains of virus cause the same pathologies. It would be interesting to investigate whether immune responses not only differ between severe and mild courses of the disease, but also between patients infected with different variants of the virus. Another limitation is the spectrum of parameters analyzed. While already covering a broad spectrum of mediators, there are many other factors which have not been investigated, including factors of the complement system, which is tightly linked to the immune and coagulation systems. Furthermore, instead of analyzing plasma and supernatants for the presence of cytokines, it would be important to analyze the expression of cytokines on a single cell basis, as the concentration measured in supernatants could be the result of a few cells producing many cytokines or many cells producing low amounts of cytokines.

## 5. Conclusions

In conclusion, after more than two years of the SARS-CoV-2 pandemic, it is widely accepted that COVID-19 is a disease with a pronounced dysregulation of immune and coagulation responses, causing organ damage. Many reports describe COVID-19 as a purely pro-inflammatory disease, requiring therapy strategies dampening a cytokine storm by e.g., glucocorticoids or anti-IL-6. In our study, we observed an increase of pro-inflammatory mediators, but at the same time immunosuppression. A similar phenotype has been described for sepsis patients. Our and findings by others imply that such patients may require immune stimulation rather than immune suppression. A similar, complex picture can be seen when looking at the coagulation system. In severe COVID-19 patients, pro-coagulation factors dominated while at the same time fibrinolysis was impaired. Consequently, due to the complexity of the disease, COVID-19 and sepsis patients might profit from personalized approaches. Analyzing a broad spectrum of immunological and coagulation parameters might help to treat individual patients according to the disease status they are currently in.

## Figures and Tables

**Figure 1 vaccines-11-00435-f001:**
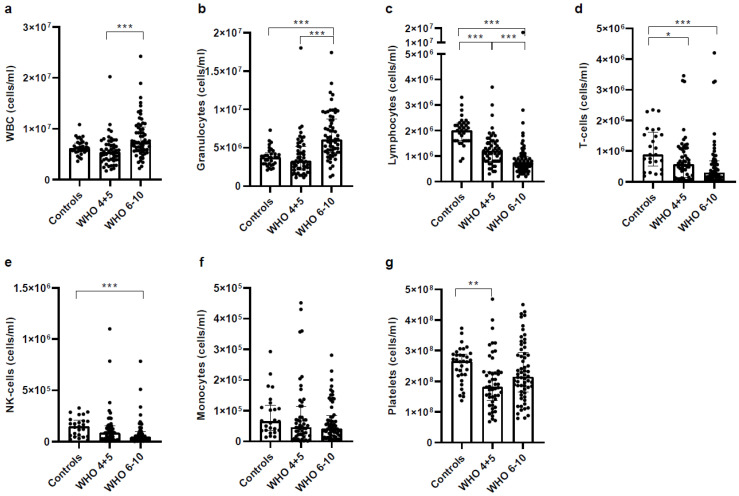
COVID-19 patients have higher granulocyte but lower lymphocyte numbers. Comparison of major immune cell populations in blood of severe COVID-19 (*n* = 75), non-severe COVID-19 patients (*n* = 52) and healthy controls (*n* = 42). Absolute numbers of major immune cell populations were determined by using a Sysmex XP300 Automated Hematology Analyzer and flow cytometry (Cytoflex S). Absolute numbers of T cells and NK cells were calculated as percentages of CD45^+^CD2^+^CD3^+^ and CD45^+^CD2^+^CD3^−^ of all leukocytes (WBCs as measured on a Sysmex XP300). (**a**) WBC, (**b**) Granulocytes (**c**) Lymphocytes (**d**) T cells (**e**) Monocytes (**f**) NK cells (**g**) Platelets. Statistical comparison between groups was done by using the Kruskal–Wallis test. Unless marked with a *p*-value, all changes observed are not significant; * *p* < 0.05; ** *p* < 0.01; *** *p* < 0.0001.

**Figure 2 vaccines-11-00435-f002:**
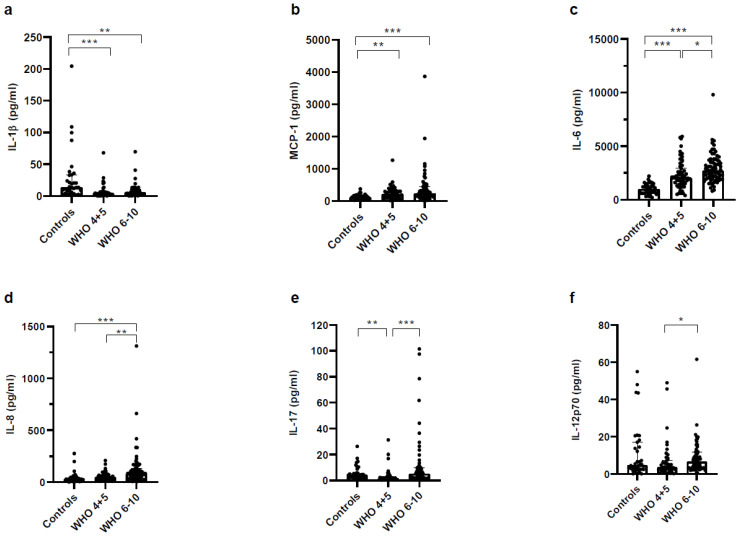
Pro-inflammatory cytokines were increased in plasma of COVID-19 patients. Platelet-free heparinized plasma of healthy controls, non-severe COVID-19 and severe COVID-19 patients was analyzed using a LegendPlex panel and measured on a Cytoflex S. Analysis was performed according to the protocol and use of software from BioLegend. Cytokines measured were (**a**) IL-1β, (**b**) MCP-1, (**c**) IL-6, (**d**) IL-8, (**e**) IL-17, (**f**) IL-12p70. Statistical comparison between groups was done by using the Kruskal–Wallis test. Unless marked with a *p*-value, all changes observed are not significant; * *p* < 0.05; ** *p* < 0.01; *** *p* < 0.0001.

**Figure 3 vaccines-11-00435-f003:**
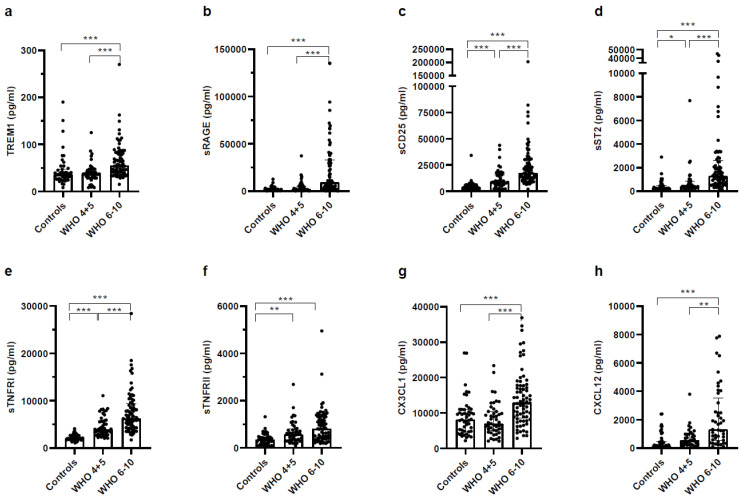
Soluble receptors and chemokines were increased in plasma of COVID-19 patients. Platelet-free heparinized plasma of healthy controls, non-severe COVID-19 and severe COVID-19 patients was analyzed using a LegendPlex panel and measured on a Cytoflex S. Analysis was performed according to the protocol and use of software from BioLegend. Mediators measured were (**a**) TREM-1, (**b**) sRAGE, (**c**) sCD25/IL-2Ra, (**d**) sST2/IL-33R, (**e**) sTNFR-I, (**f**) sTNFR-II, (**g**) CX3CL, (**h**) CXCL12. Statistical comparison between groups was done by using the Kruskal–Wallis test. Unless marked with a *p*-value, all changes observed are not significant; * *p* < 0.05; ** *p* < 0.01; *** *p* < 0.0001.

**Figure 4 vaccines-11-00435-f004:**
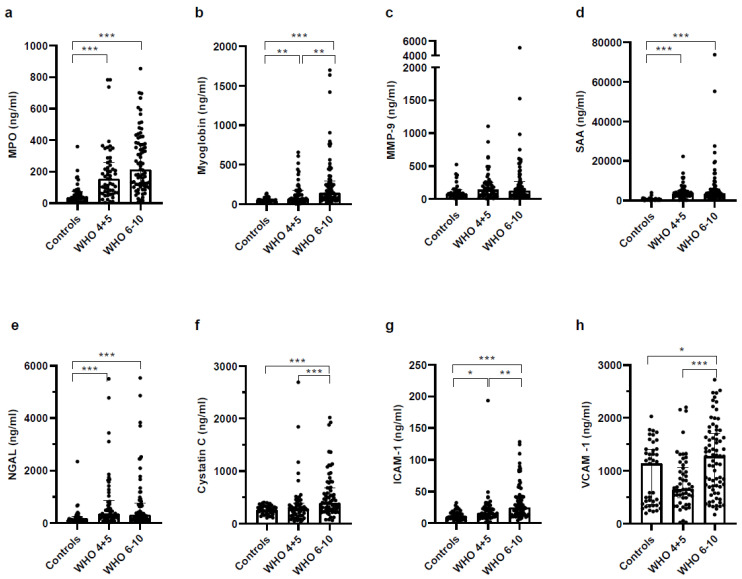
Biomarkers for organ damage and dysfunction are significantly increased in COVID-19 patients. Platelet-free heparinized plasma of healthy controls, non-severe COVID-19 and severe COVID-19 patients was analyzed using a LegendPlex panel and measured on a Cytoflex S. Analysis was performed according to the protocol and use of software from BioLegend. Mediators measured were (**a**) MPO, (**b**) Myoglobin, (**c**) MMP9, (**d**) SAA, (**e**) NGAL, (**f**) Cystatin (**g**) ICAM-1, (**h**) VCAM-1. Statistical comparison between groups was done by using the Kruskal–Wallis test. Unless marked with a *p*-value, all changes observed are not significant; * *p* < 0.05; ** *p* < 0.01; *** *p* < 0.0001.

**Figure 5 vaccines-11-00435-f005:**
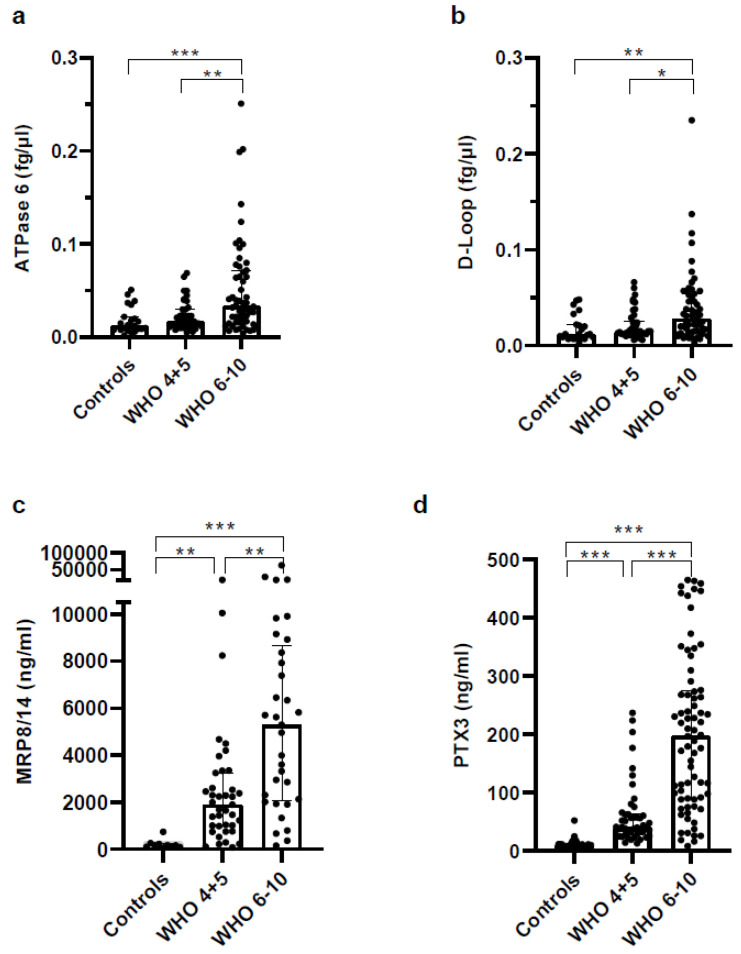
Higher levels of scavenger receptors and DAMPs are found in blood of COVID-19 patients. For analysis of mtDNA, circulating DNA was isolated from platelet-free EDTA Plasma and analyzed by RT PCR using mitochondria-specific primers. The concentration of MRP8/14 was analyzed by using selective antibodies for MRP8 and MRP14 in platelet-free EDTA plasma. PTX3 was determined in heparinized platelet-free plasma by using a LegendPlex multiplex ELISA panel. Analysis was performed according to the protocol and software provided by the companies. Mediators measured were (**a**) ATPase 6, (**b**) D-Loop, (**c**) MRP8/14/calprotectin, (**d**) PTX3. Statistical comparison between groups was done by the Kruskal–Wallis test. Unless marked with a *p*-value, all changes observed are not significant; * *p* < 0.05; ** *p* < 0.01; *** *p* < 0.0001.

**Figure 6 vaccines-11-00435-f006:**
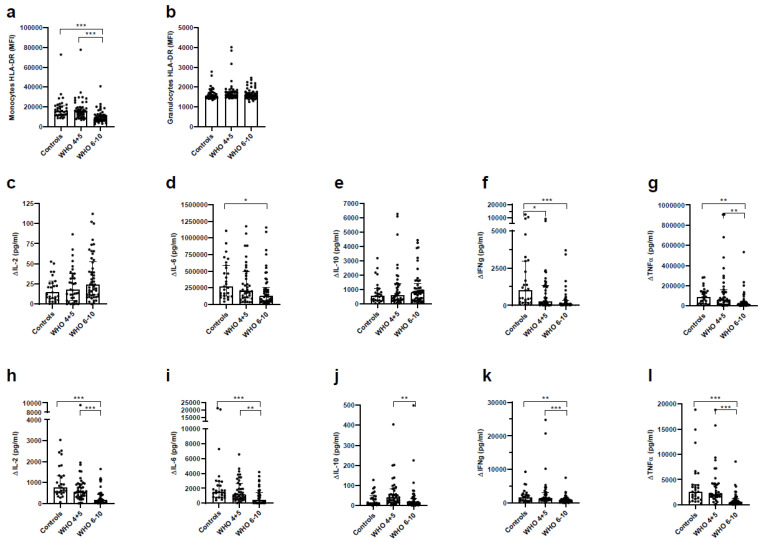
Severe COVID-19 patients show a phenotypical and functional immune suppression. Whole blood of healthy controls, non-severe COVID-19 and severe COVID-19 patients was incubated with fluorescence-labelled antibodies (CD45, CD14, CD15, HLA-DR) and measured on a Cytoflex S. Phenotypic immune suppression: Comparison of mean fluorescence intensities (MFI) of HLA-DR on (**a**) monocytes and (**b**) granulocytes. Functional immune suppression: Stimulation of (**c**–**g**) monocytes with LPS (50 ng/mL) and (**h**–**l**) of T cells (30 μL of anti-CD2CD3CD28 labelled MACSiBeads). Cytokine concentration was determined by LegendPlex analysis and normalized to 100.000 cells as determined by flow cytometry. Differences (delta) between unstimulated controls and cells stimulated with LPS and anti-CD2CD3CD28 beads, respectively, are depicted: (**c**) IL-2, (**d**) IL-6, (**e**) IL-10, (**f**) IFNγ, (**g**) TNFα and (**h**) IL-2, (**i**) IL-6, (**j**) IL-10, (**k**) IFNγ, (**l**) TNFα. Statistical comparison between groups was done by Kruskal Wallis. Unless marked with a *p*-value, all changes observed are not significant; * *p* < 0.05; ** *p* < 0.01; *** *p* < 0.0001.

**Figure 7 vaccines-11-00435-f007:**
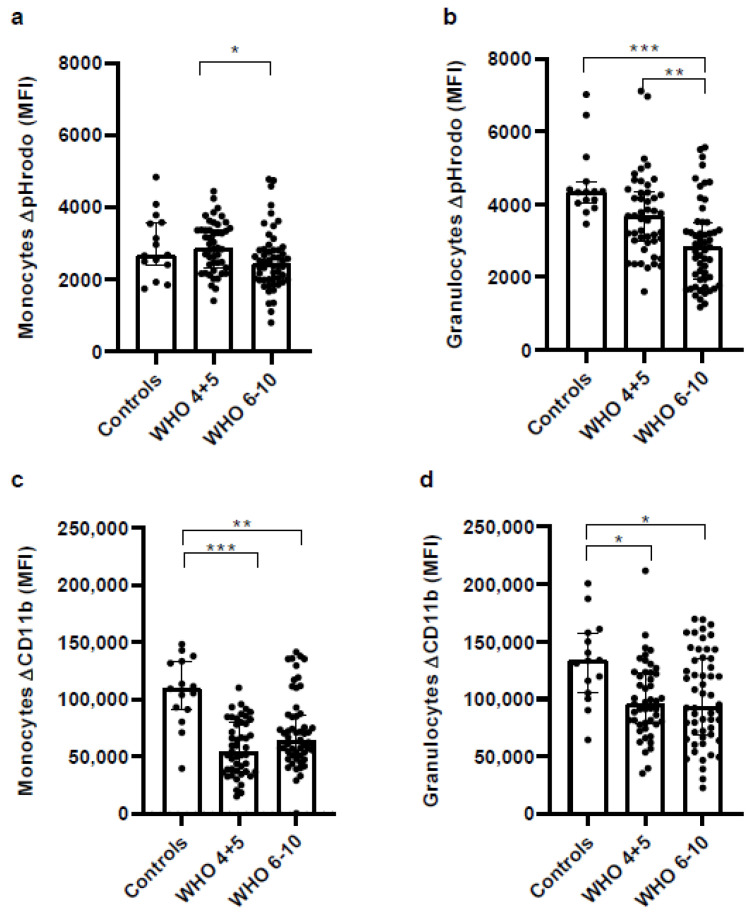
COVID-19 patients show impaired phagocytic activity of monocytes and granulocytes. Whole blood of healthy controls, non-severe COVID-19 and severe COVID-19 patients was incubated with *E.coli*-pHrodo (10μL of a 1 mg/mL solution), which is fluorescent only at low pH, i.e., after active phagocytosis. After 30 min incubation cells were analyzed for (**a**) + (**b**) phagocytosis and (**c**) + (**d**) upregulation of CD11b on (**a**) + (**c**) monocytes and (**b**) + (**d**) granulocytes. Depicted are differences (delta) of mean fluorescence intensities (MFI) between unstimulated controls and cells stimulated with *E.coli*-pHrodo. Statistical comparison between groups was done by the Kruskal–Wallis test. Unless marked with a *p*-value, all changes observed are not significant; * *p* < 0.05; ** *p* < 0.01; *** *p* < 0.0001.

**Figure 8 vaccines-11-00435-f008:**
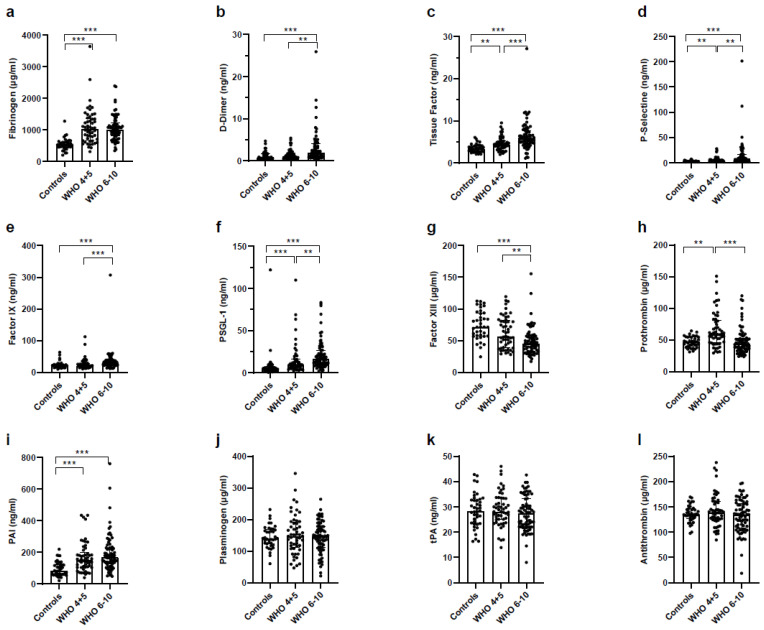
Factors involved in coagulation and fibrinolysis. Platelet-free heparinized plasma of healthy controls, non-severe COVID-19 and severe COVID-19 patients was analyzed using a LegendPlex panel and measured on a Cytoflex S. Analysis was performed according to the protocol and use of software from BioLegend. Mediators measured were (**a**) Fibrinogen, (**b**) Prothrombin, (**c**) Tissue Factor, (**d**) Factor X, (**e**) Factor XIII, (**f**) D-Dimer (**g**) Plasminogen, (**h**) Antithrombin, (**i**) tPA, (**j**) PAI-1, (**k**) PSGL-1, (**l**) P-Selectine. Statistical comparison between groups was done by using the Kruskal–Wallis test. Unless marked with a *p*-value, all changes observed are not significant; ** *p* < 0.01; *** *p* < 0.0001.

**Table 1 vaccines-11-00435-t001:** Characteristics of the study population.

Demographics	Total (*n* = 127)	WHO 4–5 (*n* = 52)	WHO 6–10 (*n* = 75)	*p*-Value
Male/Female	82/45	30/22	52/23	0.1917 ^$^
Age (years), median (range)	61 (19–88)	61 (21–88)	62 (19–86)	0.7021 ^§^
BMI (kg/m^2^), median (range)	27.8 (18.4–52.7)	25.45 (18.4–39)	29.2 (20.1–52.7)	0.0006 ^§^
Charlson Comorbidity Index, median (range)	1 (0–9)	1 (0–9)	1 (0–9)	0.1316 ^§^
SOFA on admission, median (range)	2 (0–11)	0 (0–3)	2 (0–11)	<0.0001 ^§^
SAPS II on admission, median (range)	27 (0–54)	22 (0–28)	31 (0–54)	<0.0001 ^§^
GCS, median (range)	15 (3–15)	15 (15)	15 (3–15)	0.5126 ^§^

^$^ Fisher’s test, ^§^ Mann–Whitney U test. BMI (Body Mass Index), SOFA (Sepsis-related Organ Failure Assessment), SAPSII (Simplified Acute Physiology Score II), GCS (Glasgow Coma Scale).

## Data Availability

Individual participant data cannot be made available due to EU Data Protection Regulations (GDPR). A limited and completely anonymized version of the dataset can be obtained upon request. Study protocols, including laboratory protocols will be available upon request. None of the authors was precluded from accessing data in the study, and they accept responsibility to submit for publication. Proposals should be directed to marc.berger@uk-essen.de.

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
