# Peer review of "Patients with SARS-CoV-2-Induced Viral Sepsis Simultaneously Show Immune Activation, Impaired Immune Function and a Procoagulatory Disease State"

_vaccines, 2023, doi:10.3390/vaccines11020435_

Round 1

Reviewer 1 Report

In their study, Limmer et al. investigated the immune and coagulation system status of 127 hospitalised COVID-19 patients compared to 42 healthy individuals. Regarding severe COVID-19 disease courses, the authors found evidence for the simultaneous occurrence of significantly dysregulated immune and coagulation systems. The data presented demonstrate convincingly that COVID-19 is not solely characterized by pro-inflammatory processes, but also to a certain extend by impaired innate and adaptive immunity. This is of particular interest when it comes to the application of therapies that result in lowering the patient’s immune activation. In addition, pro-coagulatory factors were shown to be enhanced with simultaneous impaired fibrinolysis. The authors highlight the importance of broadening the spectrum of markers tested to determine the patient’s immune and coagulation status to be able to apply successful individualized therapeutic strategies.

The study is well-designed and gives important insights on immunological processes during COVID-19 disease course. Even though in general, the manuscript is well written and results are presented as figures with a clear layout, the manuscript needs some revision before acceptance.

Main points of criticism:

1.       Table 1 is mentioned in the text, but not provided.

2.       Figure legend 1: n for severe COVID-19 is stated to be 76, in line 217 of the text body it is stated to be 75. Same holds for non-severe cases: n-number in the text is given as 52, but in the legend of figure 1 stated to be 52. Please clarify.

3.       There are issues regarding experimental implementation in figure 5: It is stated that RT-PCR was used to analyse mtDNA. However, this method is not mentioned in the material & methods section. An adequate paragraph has to be added to the manuscript, describing RT-PCR and primer sequences used. Furthermore, I am a bit confused as far as Figure 5b is concerned. As far as I understood, for analysis of the mt D-loop, a triple-stranded segment found in the major NCR of many mitochondrial genomes, RT-PCR was done. If so, the y-axis is not plausible, as one would expect seeing Ct-values instead of fg/µl. Please explain in more detail how you quantified mtDNA and add an adequate passage to materials & methods section.

4.       The paragraph titles of the results section are not meaningful. It is strongly recommended to introduce meaningful headliners, summarizing the findings of the following paragraph, thus supporting easy readability and intelligibility.

5.       The discussion section refers to the results found and comparisons to current literature are appropriate. However, it has to be shortened, discussing the main points of the study in a more concise manner. The same holds for the conclusion – this part needs to be shortened as well.

Minor comments and typing errors:

1.       Line 167: Space too many: SAA, …

2.       Line 185: “(all from Biolegend),).” Please remove comma and one bracket.

3.       Please add n = x in figure legend 1 (line 236). Indication of n as e.g. “(51)” could be confused with reference list numbers.

4.       Line 241: Remove sentence: “a) WBC, b) Granulocytes (…)” – repetitive, cell types are indicated as y-axis legend of figures anyway.

5.       Line 244/262/282/301/324/358/383/408/Figure legend S2: “*P<0.05, **p<0.01, ***p<0.0001” à it should be always “p” or “P”, but please do not mix it up.

6.       Line 246: “Figures 2” à Figure 2

7.       Line 258/277/403/Figure legends S1 and S2: Remove bracket after Cytoflex S.

8.       Line 259/278/298/321: remove listing of factors shown. It is repetitive and indicated in the figures anyway.

9.       Line 328: You have to be more explicit when stating “various assays” – please name the methods applied.

10.   Line 339: Space is missing: “(Fig. 6h-l).Overall”

11.   Figure 7, line 379: There is no figure e). à b) + d) (?)

12.   Line 429: different types of “-“ are used, please stick to one type within the text.

13.   Line 559: space is missing: “NLRwas”

14.   Line 561: reference to table 1 à see also 4.

15.   Line 571: one space too many: “It is not known”.

Author Response

Point to point reply to comments of reviewer #1

Dear Reviewer,

Thank you for giving us the opportunity to reply to your comments on our manuscript „Patients with SARS-CoV-2-induced viral sepsis simultaneously show immune activation, impaired immune function and a procoagulatory disease state“ which has been submitted January 14th 2023.

We want to thank you for taking the effort to carefully read and comment our manuscript. The comments were all very helpful!

Reviewer #1:

  1. Table 1 is mentioned in the text, but not provided.

Our response:

Thank you for reminding us! We added Table 1 to the manuscript

  1. Figure legend 1: n for severe COVID-19 is stated to be 76, in line 217 of the text body it is stated to be 75. Same holds for non-severe cases: n-number in the text is given as 52, but in the legend of figure 1 stated to be 52. Please clarify.

Our response:

The number of patients has been corrected. The number of severe-COVID-19 patients in the study was n=75, the number for non-severe-COVID-19 patients was n=52.

  1. There are issues regarding experimental implementation in figure 5: It is stated that RT-PCR was used to analyse mtDNA. However, this method is not mentioned in the material & methods section. An adequate paragraph has to be added to the manuscript, describing RT-PCR and primer sequences used. Furthermore, I am a bit confused as far as Figure 5b is concerned. As far as I understood, for analysis of the mt D-loop, a triple-stranded segment found in the major NCR of many mitochondrial genomes, RT-PCR was done. If so, the y-axis is not plausible, as one would expect seeing Ct-values instead of fg/µl. Please explain in more detail how you quantified mtDNA and add an adequate passage to materials & methods section.

Our response:

A detailed description of mtDNA analysis has been added to the method section. The fg/µl on the y-axis is correct, as the depicted values are absolute values and not relative ones.

  1. The paragraph titles of the results section are not meaningful. It is strongly recommended to introduce meaningful headliners, summarizing the findings of the following paragraph, thus supporting easy readability and intelligibility.

Our response:

The paragraph titels in the result section have been changed to more meaningful summaries.

  1. The discussion section refers to the results found and comparisons to current literature are appropriate. However, it has to be shortened, discussing the main points of the study in a more concise manner. The same holds for the conclusion – this part needs to be shortened as well.

Our response:

We have shortended the conclusion section by 50%.

However, although we understand the criticsm of the reviewer, we are afraid to leave important points unaddressed, if we shorten the discussion by more than a few sentences. COVID-19 is such a complex disease covering a broad spectrum of immune and coagulation phenomena. Many of our findings are new and contradictory to other findings in the field. Therefore, all kinds of different aspects found in our study have to be set into context with other findings. Furthermore, as our mansucript is supposed to be part of a special edition, we think that it is important to not only talk about the findings themselves but in additon to discuss their implications. For example, although biomarkers were not in the main focus of our study, some markers (eg MRP8/14, PTX, NLR) might function as biomarkers. By discussing these aspects we also think that the mansucript might address a broader audience.

To the minor comments and spelling errors:

  1. Line 167: Space too many: SAA, …
  2. Line 185: “(all from Biolegend),).” Please remove comma and one bracket.
  3. Please add n = x in figure legend 1 (line 236). Indication of n as e.g. “(51)” could be confused with reference list numbers.
  4. Line 241: Remove sentence: “a) WBC, b) Granulocytes (…)” – repetitive, cell types are indicated as y-axis legend of figures anyway.
  5. Line 244/262/282/301/324/358/383/408/Figure legend S2: “*P<0.05, **p<0.01, ***p<0.0001” à it should be always “p” or “P”, but please do not mix it up.
  6. Line 246: “Figures 2” à Figure 2
  7. Line 258/277/403/Figure legends S1 and S2: Remove bracket after Cytoflex S.
  8. Line 259/278/298/321: remove listing of factors shown. It is repetitive and indicated in the figures anyway.
  9. Line 328: You have to be more explicit when stating “various assays” – please name the methods applied.
  10. Line 339: Space is missing: “(Fig. 6h-l).Overall”
  11. Figure 7, line 379: There is no figure e). à b) + d) (?)
  12. Line 429: different types of “-“ are used, please stick to one type within the text.
  13. Line 559: space is missing: “NLRwas”
  14. Line 561: reference to table 1 à see also 4.
  15. Line 571: one space too many: “It is not known”.

Our response:

We are thankful for providing us with such an extensive list of mistakes we made. We corrected all mistakes you made us aware of.

Reviewer 2 Report

This study carried out by Limmer and collaborators is very interesting since it describes in particular ways aspects that are very interesting in patients with severe and non-severe SARS-CoV-2 symptoms. In this sense, the study shows robust data, but some controls are missing to give the study greater robustness.

Modify some graphs' significance bars since some overlap with the points distributed within the graphs.

Please change the signs of alpha, gamma, and all that correspond to the correct ones.

In the figures of activation of monocytes and T lymphocytes, a specific antigen control is missing, such as Spike protein or another associated with the virus.

Modify line 332 (Fig 6a) by Fig 6a -6b

It is not clear why to test for cytokines in the LT that are specific to the innate response.

Figure 7c shows how it is explained that the monocytes of the WHO 4+5 group tend to decrease the expression of CD11b compared to the WHO 6-10 group. What is the value of p?

Although the authors discuss the results, the cytokines are related only to the clinical picture. However, it would be necessary to correlate these results, and how they could affect the activity of the cells they observe affected within their evaluations.

Upload the information from lines 572 to 571

Author Response

Point to point reply to comments of eviewer #2

Dear Reviewer,

Thank you for giving us the opportunity to reply to your comments on our manuscript „Patients with SARS-CoV-2-induced viral sepsis simultaneously show immune activation, impaired immune function and a procoagulatory disease state“ which has been submitted January 14th 2023.

We want to thank you for taking the effort to carefully read and comment our manuscript. The comments were all very helpful!

To major comment 1+4:

  1. This study carried out by Limmer and collaborators is very interesting since it describes in particular ways aspects that are very interesting in patients with severe and non-severe SARS-CoV-2 symptoms. In this sense, the study shows robust data, but some controls are missing to give the study greater robustness.

Our response:

We assume that the „controls“ you comment on are the same as those addressed in comment 4, i.e. lack of virus-specific antigen controls.

In our study we did not address antigen-specific, antiviral T cell responses. We only looked at the general capacity of T cells to respond to a general stimulus, provided by stimulatory antibodies bound to beads (crosslinking the T cell receptor (TCR) and costimulatory molecules). These stimulatory beads are used in the field, if an antigen unspecifc stimulus is wanted, e.g. for expansion of T cells of unknown specificity. As we and others have previously observed that T cells in trauma or sepsis patients show a general lack of reactivity, we wanted to test whether T cells in COVID-19 patients were able to respond to a stimulus directed to their TCR.

  1. Modify some graphs' significance bars since some overlap with the points distributed within the graphs.

Our response:

We looked at all graphs again and tried to avoid overlaping of significance bars and data points. In the current form we think that no data points are hidden and therefore, the conclusions drawn by the graphs are not at risk.

  1. Please change the signs of alpha, gamma, and all that correspond to the correct ones.

Our response:

All greek letters have been changed.

  1. In the figures of activation of monocytes and T lymphocytes, a specific antigen control is missing, such as Spike protein or another associated with the virus.

Our response:

As we assume that you refer to the same „missing controls“ in comment 1 and 4, we kindly ask you to look at our response to comment 1.

  1. Modify line 332 (Fig 6a) by Fig 6a -6b

Our response:

6b has been added.

  1. It is not clear why to test for cytokines in the LT that are specific to the innate response.

Our response:

We are not completely sure, which cytokines are meant, but we think you refer to Figure 6, where we stimulated macrophages with LPS and T cells with anti-CD2CD3CD28 beads. The cytokines analyzed were chosen because ALL of them are indeed cytokines produced by T cells. Most of the cytokines (besides IL-2) are also produced by cells of the innate immune system, such as macrophages. Some cytokines are mainly produced by macrophages (IL-6, TNF), but can also be secreted by T cells. For a detailed analysis of e.g. IL-6 producing T cells, cells should be analyzed by intracellular cytokine staining (ICS). ELISA data only give cumulative data of all cells in the well, while ICS data allow the analysis of individual cells. We cannot differentiate between many cells secreting low amounts of cytokines and a few cells secreting enormous amounts of cytokines.

To major comment 7:

  1. Figure 7c shows how it is explained that the monocytes of the WHO 4+5 group tend to decrease the expression of CD11b compared to the WHO 6-10 group. What is the value of p?

Our response:

The difference between WHO 4+5 and WHO 6-10 is not significant, despite the tendency you described. To prevent overloading graphs with „ns“ (non significant) marks, we only included * marks representing significant differences. To avoid any misunderstandings we wrote in all figure legends that changes without a p-value are not significant.

To major comment 8:

  1. Although the authors discuss the results, the cytokines are related only to the clinical picture. However, it would be necessary to correlate these results, and how they could affect the activity of the cells they observe affected within their evaluations.

Our response:

It is very difficult to discuss cause and consequences of cytokines for the clinical picture. Our observation that inflammatory and anti-inflammatory cytokines are elevated simultaneously makes things even more complicated. To some respect our data even contradict some generally accepted conclusions, as e.g. that patients with an elevated IL-6 level should be treated with anti-IL-6 reagents. Our finding that inflammatory and anti-inflammatory cytokines are elevated at the same time and in addition immune functions are impaired is rather supporting immune stimulatory therapies. However, based on current knowledge, conclusions on the effect of certain cytokines on individual cells are not possible. This would be speculation. Hopefully future research will help to clarify those aspects.

  1. Upload the information from lines 572 to 571

Our response:

Has been corrected

Reviewer 3 Report

The researchers investigated immune responses in COVID-19 patients, and found that some immune responses were active., along with coagulation processes, while other immune responses were inhibited. Some comments addressing this issue are included in the attached pdf file. 

Author Response

Point to point reply to comments of reviewer #3

Dear Reviewer,

Thank you for giving us the opportunity to reply to your comments on our manuscript „Patients with SARS-CoV-2-induced viral sepsis simultaneously show immune activation, impaired immune function and a procoagulatory disease state“ which has been submitted January 14th 2023.

We want to thank you for taking the effort to carefully read and comment our manuscript. The comments were all very helpful!

  1. For future research, you may want to consider dietary sodium chloride overload as a contributing factor which, in addition to causing coagulopathy, causes an immune response while inhibiting mucociliary clearance in the nasopharyngeal mucosal immune system that may lead to sepsis: https://www.mdpi.com/1648-9144/57/8/739

We thank the reviewer for suggesting to look at dietary sodium chloride overload as contributing factor to organ damage in the lung. The connection between immune system und coagulation system is already very complex and it is therefore conceivable that factors impairing normal mucosal immune responses have a negative effect on overall lung functions. We will consider this aspect for future analyses.

  1. Are these causes or consequences of some other causative agent?

Our response:

As in the introduction we discuss current literature we can only refer to the conclusions of other authors. In those reviews cited the authors concluded that the observed organ damage was the consequence of the virus infection (direct damage to infected tissue) and the subsequent massive inflammation (indirect immune damage due to mediators secreted by granulocytes, macrophages, mast cells etc.). Due to the systemic character of many secreted mediators the authors assumed that further damage was the result of these systemically spread cytokines and chemokines.

Although we do not want to exclude the effect of other causative agent, we think that the virus is responsible for most of the direct and indirect damage observed.

  1. What about pulmonary edema that causes shortness of breath and death? https://www.frontiersin.org/articles/10.3389/fphar.2021.664349/full#:~:text=COVID%2D19%20mortality%20is%20primarily,%2DCoV%2D2)%20infection.

Our response:

The aspect of pulmonary edema is an important point. As we we did not want to refer to a single aspect and thus wrote „lung damage“ to cover a broad spectrum. However, as we consider this point to be important, we would like to add the reference mentioned by the reviewer: New reference 16: Cui X, Chen W, Zhou H, Gong Y, Zhu B, Lv X, et al. Pulmonary Edema in COVID-19 Patients: Mechanisms and Treatment Potential. Front Pharmacol. 2021;12:664349.

As the manuscript in its current format does not allow us to add references via Endnote, we would kindly ask the editor to add the reference to our manuscript.

Round 2

Reviewer 1 Report

Dear authors,

no further comments, only another typing error:

line 245: increassed >increased

all the best and kind regards,

the reviewer